Effect of thermal cycling on the mechanical properties of conventional, milled, and 3D-printed base resin materials: a comparative in vitro study

Xiao Shuang 1 2 3
Zhang Ruo-Jin 1 2
Tan Fa-Bing 1 2 4 xiaosongtan1983@hospital.cqmu.edu.cn
1 College of Stomatology, Chongqing Medical University , Chongqing , China
2 Chongqing Key Laboratory of Oral Diseases and Biomedical Sciences , Chongqing , China
3 Chongqing Medical University Experimental Teaching Management Center, Chongqing Medical University , Chongqing , China
4 Chongqing Municipal Key Laboratory of Oral Biomedical Engineering of Higher Education , Chongqing , China
Mourão Carlos Fernando
Electronic publication date: 2025 Mar 17
Publication date: 2025
Volume: 13
Electronic Location ID: e19141
Received 2024 Dec 11; Accepted 2025 Feb 19
Copyright: © 2025 Xiao et al.
Copyright year: 2025
Copyright holder: Xiao et al.
License: This is an open access article distributed under the terms of the Creative Commons Attribution License, which permits unrestricted use, distribution, reproduction and adaptation in any medium and for any purpose provided that it is properly attributed. For attribution, the original author(s), title, publication source (PeerJ) and either DOI or URL of the article must be cited.
License URL: https://creativecommons.org/licenses/by/4.0/

Keywords: Denture base materials, Milled, 3D-printed, Thermal cycling, Mechanical properties

Funding: Joint Medical Research Project between Chongqing Science and Technology Bureau and Health Commission 2023MSXM094 Chinese Medical Association and the National Center for the Development of Medical Education 2023A50 Chongqing College Student Innovation and Entrepreneurship Training Program S202310631017 This work was supported by the Joint Medical Research Project between Chongqing Science and Technology Bureau and Health Commission (No. 2023MSXM094), the 2023 medical education research project of the Medical Education Branch of the Chinese Medical Association and the National Center for the Development of Medical Education (No. 2023A50) and the Chongqing College Student Innovation and Entrepreneurship Training Program (No. S202310631017). The funders had no role in study design, data collection and analysis, decision to publish, or preparation of the manuscript.

==============================
Purpose

The purpose of this study is to evaluate the impact of thermal cycling on the mechanical properties of conventional, milled, and 3D-printed denture base materials.

Methods

Unigraphics NX software was used to design the sample data, after which denture base resin samples were fabricated using conventional polymerization (conventional), milling, and 3D-printing techniques. Flexural strength, Vickers hardness, and impact strength of each group of samples were evaluated both before and after 10,000 thermal cycles in distilled water at 5 °C and 55 °C (n = 8/group). Statistical analysis of the data was conducted using the Kruskal-Wallis H test, Weibull analysis and Spearman correlation analysis.

Results

The flexural strength and impact strength of the 3D-printed group significantly decreased after thermal cycling (P = 0.001), whereas no significant differences were observed before and after thermal cycling in the conventional or milled groups (P > 0.05). No significant correlation was found between flexural strength and impact strength for any of the groups. The Weibull modulus of 3D printed groups for both flexural and impact strength decreased after thermal cycling. The Vickers hardness of the conventional group increased significantly after thermal cycling, while Vickers hardness significantly decreased in the milled or 3D-printed groups (P < 0.05).

Conclusion

Compared with the conventional or milled groups, thermal cycling had a more pronounced effect on the flexural strength, Vickers hardness, and impact strength of the 3D-printed group. These findings indicate that further improvements (e.g., material composition, printing parameters and post-processing) in the mechanical properties of 3D-printed materials is necessary before clinical application.

Introduction

Polymethyl methacrylate (PMMA) has become the primary material for denture base fabrication due to its favorable color stability, lightweight nature, and cost-effectiveness (Al-Fouzan, Al-Mejrad & Albarrag, 2017). For several decades, conventional heat polymerization of PMMA has been the predominant method for denture base fabrication. However, this method has limitations, including a complex process, lengthy time requirements, and inconsistent quality. With technological advancements, digital processing techniques such as milling and 3D-printing have emerged as promising alternatives to enhance the quality and efficiency of denture base fabrication. Milling can significantly increase the degree of resin polymerization and reduce residual monomers by processing PMMA resin discs under high temperature and pressure (Arslan et al., 2018). 3D printing, in contrast, allows for the simultaneous production of multiple denture bases, reducing material waste by up to 40% (Greil et al., 2023). This approach not only enhances production efficiency but also holds substantial potential for broader clinical application.

Denture base materials are widely recognized to undergo thousands of occlusal forces daily, with pressures reaching up to 110 N (Prombonas & Vlissidis, 2006). Multi-directional and complex masticatory loads readily contribute to denture fractures or failure. Consequently, superior mechanical properties are essential to ensure the functionality and durability of denture bases. The mechanical properties of denture base resins include flexural strength, hardness (e.g., Vickers hardness), impact strength, and other key indicators.

Flexural strength is a primary indicator of a denture base resin’s capacity to withstand masticatory stress, making it crucial for evaluating the material’s mechanical properties. Excessive occlusal forces or uneven occlusion can substantially elevate flexural stress (Hamdy, 2024). When this stress exceeds the resin’s maximum load-bearing capacity, the denture base is susceptible to deformation or fracture, thereby compromising its long-term usability. Hardness is another critical property, reflecting the material’s resistance to indentation or scratching under a specified load. A low hardness in base resin can accelerate the denture’s wear rate during prolonged chewing (Altarazi et al., 2024), increasing the risk of material fracture and adversely affecting patient experience. Impact strength measures a material’s resistance to fracture under sudden impact, which is crucial for the overall resilience of the base resin to transient stress or impact forces. Research indicates that approximately 80% of mandibular denture fractures are caused by impact forces (Sasaki et al., 2016). Without adequate impact strength, denture base resins may crack or sustain damage due to accidental drops or unexpected events. In sum, the mechanical properties of denture base resins are essential to the longevity and reliability of dentures.

The intraoral environment functions as a thermodynamic medium, with temperature fluctuations caused by ingested food and beverages. One study (Ayaz, Bağış & Turgut, 2015) reported that oral temperature fluctuates between 20 and 50 times per day, averaging 10,000 fluctuations annually. Extended exposure of denture base resins to humid conditions enables water molecules, which act as plasticizers, to progressively infiltrate the polymer network, reducing intermolecular forces (Machado et al., 2012). Currently, artificial thermal cycling is commonly employed to simulate intraoral temperature changes, serving as an effective approach for evaluating the long-term durability of denture bases (Abdul-Monem & Hanno, 2024; Çakmak et al., 2024). Polychronakis et al. (2017) evaluated the impact of thermal cycling on the flexural strength of conventional base resins, noting a significant decrease in flexural strength in conventional PMMA after 3,000 and 5,000 thermal cycles. Similarly, Atalay et al. (2021) investigated changes in the hardness of CAD-CAM denture base materials with various surface treatments following thermal cycling, finding a reduction in Knoop hardness across all PMMA groups.

Nonetheless, studies on the mechanical properties of 3D-printed denture base resins remain limited. Some studies (Alshali et al., 2024; Viotto et al., 2022; Casucci et al., 2023; Zeidan et al., 2023) have assessed only certain mechanical properties (e.g., flexural strength), while others (Prpić et al., 2020; Souza et al., 2024; Arora et al., 2024; Dwivedi et al., 2024; Lee et al., 2023) have analyzed multiple indicators without simulating the oral environment through thermal cycling. Moreover, findings in this area have been inconsistent. For instance, El Samahy et al. (2023) observed a significant decrease in the flexural strength of 3D-printed denture base resins after thermal cycling, whereas Altarazi et al. (2024) reported no significant change in flexural strength after artificial aging. Such limited or conflicting results have raised numerous questions regarding the clinical mechanical properties of 3D-printed base resins.

To thoroughly and rigorously evaluate the mechanical properties of newly developed 3D-printed base resins, this study assessed the effects of simulated oral thermal cycling conditions (10,000 cycles in distilled water at 5 °C and 55 °C) on the flexural strength, Vickers hardness, and impact strength of conventional, milled, and 3D-printed base resin materials, with conventional base resin serving as a control. This study aids in evaluating the reliability and longevity of base resin applications in the oral environment using different manufacturing methods, providing a theoretical basis and technical support for future material selection and process optimization. The initial hypothesis posited that thermal cycling would have no differential effect on the mechanical properties of conventional, milled, and 3D-printed denture resin materials.

Materials and Methods

Constructing sample data

Following ISO 20795-1:2013 (2013), Unigraphics NX10.0 software (Siemens PLM Software, Plano, USA) is used to precisely design the rectangular samples of 65 mm × 10 mm × 3 mm for three-point flexural strength test. According to ISO 179-1:2023 (2023), samples measuring 50 mm × 6 mm × 4 mm were designated for the impact strength test. These designs were saved in STL file format. Sample size was calculated using G*Power 3.1.9 software (program written by Franz-Faul, University of Düsseldorf, Germany) with an effect size of 0.3, a statistical power (1-β) of 0.9, and a significance level of 0.05. The required sample size for each group was determined to be 7.83; consequently, the sample size was increased to eight per group in this study. The materials used here are the same as in the previous study (Zhang et al., 2024), as detailed in Table 1.

Table 1 Three types of resin base materials used in this study.

Material	Fabrication technique	Composition	Production information	
Denture base polymer	Conventional	Powder: polymethyl methacrylate, pigments
Liquid: methyl methacrylate	Lot: 230603
Manufacturer: Shanghai Pigeon Dental MFG Co. Ltd.	
Denture base resin discs	Milled	Polymethyl methacrylate, Methyl methacrylate, Ethylene glycol dimethacrylate, Pigments	Lot: 200818010
Manufacturer: Shandong Huge Dental Material Corporation	
Fluid resins for 3D printing	3D-Printed	Di-2-methylpropanoic acid acyloxyethyl-2,2,4-trimethylhexane di-carbamate, 1,6-Hexanediyl bis(2-methylacrylate), Diacrylic acid, diester with, 3’- (isopropylidene)bis(p-phenyleneoxy)di(propane-1,2- diol), Strontium glass powder, Barium glass powder, Silicon powder, The others(Camphorquinone, 4-Methoxyphenol, Ferric oxide)	Lot: YT2-231122
Manufacturer: SinoDentex Co. Ltd.	

Conventional sample preparation

The STL data of the rectangular samples were initially imported into a 5-axis milling machine (Ideal Mill 5A; Sino-Digital Co., Ltd, Beijing, China) to mill the PMMA disk (7#A3-PINK; Shandong Huge Dental Material Corporation, Shandong, China) for producing base resin samples. Next, the samples were transformed into a silicone rubber negative impression (73221009; Shandong Huge Dental Material Corporation, Shandong, China), which was subsequently filled with molten wax to create wax samples. After applying two layers of Vaseline, the wax samples were embedded in dental stone (E.50102; Heraeus Kulzer GmbH, Hanau, Germany) under vacuum conditions. Upon opening the casts, the wax was removed using a boiling water gun to seal the plaster cavities. Denture base polymer powder and liquid (230603; Shanghai Pigeon Dental MFG Co. Ltd, Shanghai, China) were mixed in a 100 g to 45–48 ml ratio according to the manufacturer’s instructions. During the dough stage, the mixture was filled and pressed into a plaster cast and placed in a constant-temperature water bath at 70–75 °C for 1.5–2 h. The temperature was then gradually increased to boiling and maintained for 30–60 min before the process was halted. The cast was opened, and the samples were removed after cooling naturally. A mechanical grinder (ES50T-HR, Songben Co., Ltd, Guangdong, China) was used to remove any burrs from the sample surfaces. Ultimately, 32 samples with intact surfaces were obtained, with 16 designated for three-point flexural tests before and after thermal cycling, and 16 allocated for impact strength tests before and after thermal cycling, with eight samples in each group.

Milled samples

The STL data of the rectangular samples were imported into a CAM programming system (hyperDENT Compact, Version 9.4.2. GeoMedi Corporation, Berlin, Germany) to select the appropriate milling equipment and material, thereby generating the NC machining program. This program was subsequently loaded into a 5-axis milling machine (Ideal Mill 5A; Sino-Digital Co., Ltd, Beijing, China), where a resin disc (7#A3-PINK; Shandong Huge Dental Material Corporation, Shandong, China) was clamped onto the machine fixture. The machining parameters were carefully set as follows: ball-end tools of sizes R1.0*RC16*D40*TL50 (P01) and R0.5*RC16*D40*TL50 (P02) were selected. The processing strategies included P01 for rough processing (P = 22,000 rpm, F = 1,500 mm/min) and P02 for fine processing (S = 25,000 rpm, F = 1,000 mm/min). A semi-circular disk fixture was chosen, with resin disk specifications as follows: thickness = 35 mm, diameter = 98.5 mm. The upper and lower surfaces of the samples were oriented perpendicularly to the disk plane, and the processing mode was set to automatic. After processing, the samples were detached from the disk using a tungsten carbide acrylic drill, and surface dust was removed with a brush. The sample numbers and grouping were identical to those of the conventional method.

3D-printed samples

The STL data of the rectangular samples were imported into a 3D printer (IBEE300; UNIZ Technology LLC, Beijing, China), and the liquid resin material (YT2-231122; Sino-Dentex Co., Ltd, Jilin, China) was printed into samples according to the manufacturer’s instructions. The printing parameters were carefully set: LED light source wavelength of 405 nm and print layer thickness of 50 μm. The construction angle was adjusted such that the plane formed by the length and width of the samples was perpendicular to the structural plate plane, while the support structure was attached to the plane formed by the width and height of the denture base resin. Following previous methodologies (Yan et al., 2024), excess resin was removed with a 90% isopropyl alcohol solution, and the support structure was detached using silicon carbide sandpaper after printing. The samples were then cured in a light-curing oven (UV OVEN, Prismlab China Co., Ltd, Shanghai, China) at a wavelength of 405 nm for 5 min. The number and grouping of samples were consistent with those used in the conventional and milling methods.

The three groups of processed samples are shown in Fig. 1. Based on previous research methods (Çakmak et al., 2022), each group of samples was sequentially wet-ground using 800 mesh, 1,200 mesh, and 2,000 mesh silicon carbide sandpapers (3M401Q, Zhuhai Zhongli Trading Co., Ltd, Guangdong, China) until the support attachment surface was flat and smooth. All samples were then ultrasonically cleaned with distilled water for 5 min. Finally, the dimensional consistency of the samples was verified with digital calipers, ensuring dimensional variations did not exceed 0.2 mm.

Figure 1 Partially processed base resin samples.

(A) Conventional. (B) Milled. (C) 3D-printed.

Thermal cycling

Each group of samples was subjected to 10,000 thermal cycles in a high and low-temperature exchanger (TC-501FIII, Suzhou Will Experimental Supplies Co., Ltd, Suzhou, China) in a distilled water bath at 5 °C and 55 °C. The process involved immersing the samples in the low-temperature bath for 30 s, lifting them to the liquid surface, and subsequently immersing them in the high-temperature bath, with an interval of 15 s. The samples were then left in the high-temperature bath for an additional 30 s before being lifted again, completing one cycle. The rotation arm of the device facilitated the exchange of samples between baths and immersion in the water.

Mechanical property testing

The flexural strength, Vickers hardness, and impact strength of the three groups were evaluated both before and after thermal cycling. The testing process for each mechanical property is illustrated in Fig. 2.

Figure 2 Testing process of mechanical properties of the three groups of base resin samples.

(A) Three point flexural test. (B) Vickers hardness test. (C) Impact strength test.

Flexural strength test

Each group was subjected to a three-point flexural test using a universal mechanical testing machine (C43.104; MTS Inc., Eden Prairie, MN, USA). Following a previous study (Falahchai et al., 2023), the span was set to 50 mm, and the loading force was uniformly increased from 0 at a constant rate of 5 mm/min until fracture occurred. The load-deflection curves were recorded by a connected computer. Flexural strength was calculated using the following equation:

FS=3FL/(2bd2).

In the formula, FS represents the flexural strength (MPa), F denotes the maximum load or force (N) at which fracture occurs, L is the span between brackets (mm), b is the width (mm), and d is the thickness (mm).

Vickers hardness test

Following a previous study (Zeidan et al., 2022), the fractured samples from the three-point flexural test were placed on a Vickers hardness testing machine (HVS-1000; Shaoxing Jingbo Testing Instrument Co., Ltd, Zhejiang, China). The test parameters included a dwell time of 30 s and a load of 25 gf. Each sample was randomly measured at three different positions, and the average of these three readings was recorded as the hardness value for each sample. Hardness values were expressed in units of HV.

Impact strength test

According to the ISO 179-1:2023 (2023), each unnotched sample was positioned horizontally close to the support, with the span set to 40 mm. The impact testing was performed using an impact strength testing machine (XJC-50D2; Chengde Precision Testing Machine Co., Ltd, Hebei, China) equipped with a 4 J pendulum. The strike center of the hammer blade was aligned with the sample’s lengthwise and thicknesswise center. The energy absorbed by the fracture of the calibrated samples (Ec) was recorded, and the impact strength was calculated using the following equation:

I=Ec/WT.

In the formula, I represents the calculated impact strength (kJ/m²), W is the width (m), and T is the thickness (m).

SEM observation

Following a previous study (Al-Dwairi, Al Haj Ebrahim & Baba, 2023), the impact strength test, one fractured sample from each group was randomly selected after the impact test for gold sputtering using ion sputtering equipment (MC1000; Hitachi High-Tech Co., Ltd, Tokyo, Japan). The fracture surface morphology of these samples was subsequently examined using a scanning electron microscope (SEM) (SU8010; Hitachi High-Tech Co., Ltd, Tokyo, Japan) at an accelerating voltage of 5 KV and a magnification of 1,000×.

Statistical analysis

SPSS software (SPSS Statistics for Windows, Version 27.0. Armonk, NY: IBM Corp.) was utilized for the statistical analysis. The Shapiro-Wilk test was used to assess normality, and the Levene test was employed to examine the homogeneity of variances. Data that conformed to normality and homogeneity of variances were analyzed with the ANOVA test, while data that did not meet these assumptions were analyzed using the Kruskal-Wallis H test. Additionally, a two-parameter Weibull cumulative distribution was conducted on the flexural strength and impact strength (Origin(Pro), Version 2024; OriginLab Corporation, Northampton, MA, USA.) to calculate the Weibull modulus and Scale parameter. The correlation between flexural strength and impact strength was examined using Spearman correlation analysis, with a significance level set at α = 0.05.

Results

Flexural strength and weibull analysis

Table 2 presents the flexural strength of the three groups before and after 10,000 thermal cycles. Before thermal cycling, the flexural strengths were ranked as follows: conventional group < milled group = 3D-printed group. After thermal cycling, the ranking changed to: conventional group = 3D-printed group < milled group. The flexural strength of the 3D-printed group decreased significantly after thermal cycling (P = 0.001), while no significant change was observed in the flexural strength of the conventional and milled groups before and after thermal cycling (P > 0.05).

Table 2 Flexural strength of samples before and after thermal cycles (n = 8, MPa).

The data is normally distributed with unequal variances, presented as “mean ± standard deviation.” Different superscript uppercase letters indicate significant differences (P < 0.05) before and after thermal cycling for each group (in the column direction). Different superscript lowercase letters indicate significant differences (P < 0.05) among the three groups of base resins either before or after thermal cycling (in the row direction).

	Conventional	Milled	3D-Printed	H	P	
Before	45.95 ± 7.78Ab	78.25 ± 10.75Aa	87.45 ± 9.87Aa	16.980	<0.001	
After	37.64 ± 2.26Ab	88.18 ± 2.73Aa	37.38 ± 20.56Bb	15.680	<0.001	
H	3.188	3.574	10.599			
P	0.074	0.059	0.001			

The results of the Weibull analysis for flexural strength before and after thermal cycling are presented in Fig. 3. Before thermal cycling, the 3D printed group had the highest Weibull modulus, while the conventional group had the lowest Weibull modulus. After thermal cycling, the Weibull modulus of the 3D printed group decreased, whereas the modulus of the conventional and milled group increased.

Figure 3 Weibull plots for flexural strength with 95% confidence intervals.

The Weibull modulus (m) was the upward gradient of the line. The characteristic strength (σ0) was the strength at a failure probability of approximately 63.2%.

Vickers hardness

Figure 4 depicts the Vickers hardness indentations for the three groups before and after 10,000 thermal cycles. Prior to thermal cycling, the surface of the conventional group appeared smooth, with deep indentations and slightly concave edges (Fig. 4A1). The milled group exhibited a dense linear stripe pattern with fuzzy indentation edges (Fig. 4A2). The 3D-printed group displayed a uniform linear arching pattern with rounded indentation edges (Fig. 4A3). After thermal cycling, the conventional group showed straight indentation edges (Fig. 4B1), while the milled group exhibited unclear indentation edges (Fig. 4B2), and the indentation shape of the 3D-printed group became irregular (Fig. 4B3).

Figure 4 Vickers hardness indentation before and after thermal cycling of the three groups.

Before thermal cycling: (A1, A2, A3). After thermal cycling: (B1, B2, B3).

Table 3 shows the Vickers hardness values of the three groups before and after 10,000 thermal cycles. Prior to thermal cycling, the hardness ranking was: conventional group < milled group = 3D-printed group. Following thermal cycling, the ranking shifted to: conventional group = 3D-printed group < milled group. The Vickers hardness of the conventional group increased significantly after thermal cycling, whereas the hardness values for the milled and 3D-printed groups decreased significantly (P < 0.05).

Table 3 Vickers hardness of samples before and after thermal cycles (n = 8, HV).

The data is normally distributed with unequal variances, presented as “mean ± standard deviation.” Different superscript uppercase letters indicate significant differences (P < 0.05) before and after thermal cycling for each group (in the column direction). Different superscript lowercase letters indicate significant differences (P < 0.05) among the three groups of base resins either before or after thermal cycling (in the row direction).

	Conventional	Milled	3D-Printed	H	P	
Before	14.98 ± 0.53Bb	20.31 ± 1.18Aa	17.78 ± 1.57Aab	18.060	<0.001	
After	17.06 ± 0.86Ab	19.01 ± 0.35Ba	15.68 ± 1.57Bb	15.982	<0.001	
H	10.976	8.378	6.362			
P	<0.001	0.004	0.027			

Impact strength and weibull analysis

Table 4 presents the impact strength of the three groups before and after 10,000 thermal cycles. Before thermal cycling, the impact strength of the 3D-printed group was significantly lower than that of the milled group (P < 0.001), while no significant difference was observed between the conventional group and either the milled or 3D-printed groups (P > 0.05). After thermal cycling, the impact strength hierarchy was as follows: 3D-printed group < conventional group = milled group. The impact strength of the 3D-printed group was significantly reduced after thermal cycling (P = 0.001), whereas no significant change was observed in the impact strengths of the conventional and milled groups before and after thermal cycling (P > 0.05).

Table 4 Impact strength of samples before and after thermal cycles (n = 8, KJ/m2).

The data is normally distributed with unequal variances, presented as “mean ± standard deviation.” Different superscript uppercase letters indicate significant differences (P < 0.05) before and after thermal cycling for each group (in the column direction). Different superscript lowercase letters indicate significant differences (P < 0.05) among the three groups of base resins either before or after thermal cycling (in the row direction).

	Conventional	Milled	3D -Printed	H	P	
Before	6.67 ± 1.86Aab	11.06 ± 2.34Aa	4.57 ± 0.81Ab	17.276	<0.001	
After	8.37 ± 2.97Aa	9.64 ± 2.76Aa	2.43 ± 1.67Bb	14.921	<0.001	
H	1.591	0.706	6.914			
P	0.207	0.401	0.009			

The Weibull analysis results for impact strength before and after thermal cycling are presented in Fig. 5. Before cycling, the 3D printed group had the highest Weibull modulus, and the conventional group had the lowest modulus. After cycling, all groups decreased in Weibull modulus, with the 3D printed group having the lowest modulus.

Figure 5 Weibull plots for impact strength with 95% confidence intervals.

The Weibull modulus (m) was the upward gradient of the line. The characteristic strength (σ0) was the strength at a failure probability of approximately 63.2%.

SEM observation

Figure 6 shows representative SEM images of the fracture surfaces for the three groups after impact testing (1,000× magnification). Before thermal cycling, the conventional group (Fig. 6A1) exhibited a stepped, coarse ratchet structure, while the milled group (Fig. 6A2) displayed more elongated ratchets. The 3D-printed group (Fig. 6A3) exhibited flaky, concave-like structures of various sizes, with localized micro-cracks and defects, such as small bubbles. After thermal cycling, the stepped ratchets in the conventional group (Fig. 6B1) became more prominent with curved extension traces, the bifurcated ratchets in the milled group (Fig. 6B2) increased, forming river-like patterns, and the concave-like structures in the 3D-printed group (Fig. 6B3) became less pronounced, with visible cracks and larger bubble-like defects.

Figure 6 SEM images at 1,000× magnification of typical samples before and after 10,000 thermal cycles in the three groups.

Before thermal cycling: (A1, A2, A3). After thermal cycling: (B1, B2, B3). Blue arrows indicate the ratchets, yellow arrows indicate the concave-like structures, red arrows indicate the cracks, and white arrows indicate the defects.

Correlation between flexural strength and impact strength

Table 5 displays the Spearman correlation coefficients (ρ) and the corresponding P-values for the flexural strength and impact strength of the three groups before and after 10,000 thermal cycles. The results indicated no significant correlation between flexural strength and impact strength, both before and after thermal cycling.

Table 5 Correlation coefficients between flexural strength and impact strength for three groups before and after thermal cycles.

	Conventional	Milled	3D-Printed	Total	
Before	ρ	−0.43	−0.09	−0.52	−0.24	
P	0.29	0.82	0.19	0.26	
After	ρ	−0.42	−0.17	−0.12	0.41	
P	0.30	0.69	0.78	0.04	

Discussion

This study simulated thermal cycling conditions reflective of the oral environment to assess the effect on the mechanical properties of conventional, milled, and 3D-printed base resins. The results showed that before and after thermal cycling, significant differences existed in flexural strength, Vickers hardness, and impact strength among the conventional, milled, and 3D-printed groups (P < 0.001). After thermal cycling, the 3D-printed group exhibited a greater reduction in flexural strength, Vickers hardness, and impact strength compared to the conventional and milled groups. Thus, the hypothesis was rejected.

Although both the three-point flexural test and the four-point flexural test are commonly employed methods for measuring the flexural strength of base resins, Chitchumnong, Brooks & Stafford (1989) reported that the results of the three-point flexural test were statistically more reliable when comparing the two methods for measuring the flexural strength of various polymers. Consequently, the three-point flexural test was utilized in this study. The findings indicated that the flexural strength of the milled and 3D-printed groups was significantly higher than that of the conventional group before thermal cycling. This result is consistent with a recent study demonstrating higher flexural strength in CAD/CAM-milled and 3D-printed denture base resins compared to conventional denture base materials (Temizci & Bozoğulları, 2024). The flexural strength was influenced by the degree of resin polymerization and was positively correlated with the monomer conversion rate (Aguirre et al., 2020). The milled PMMA base resin used in this study was pre-polymerized under high temperature and pressure, resulting in a longer reaction chain, a denser structure, and a higher degree of monomer conversion (Atalay et al., 2021). In contrast, the 3D-printed group was cured layer by layer according to the sheet-milling algorithm, forming a tighter interlayer bond (Temizci & Bozoğulları, 2024). When subjected to flexural load, the printed layers can work together to resist external forces, which may explain the higher flexural strength observed. After thermal cycling, the flexural strength of the base resin in the 3D-printed group decreased significantly, a result consistent with the findings of Alshali et al. (2024) and El Samahy et al. (2023). This decrease may be related to the higher water absorption capacity of the material. Water molecules not only penetrated the tiny pores within the 3D-printed base resin, causing damage to the network structure (Belirli et al., 2020), but also hydrolyzed the silane coupling agent, breaking the chemical bond between the coupling agent and the resin matrix or filler, thereby reducing the flexural strength of the resin base (Nam et al., 2021). In this study, compared to the conventional and milled groups, the SEM images of the fracture surface of the 3D-printed group revealed obvious cracks after thermal cycling, suggesting potential deeper structural damage. According to the ISO 20795-1:2013 (2013), the flexural strength of denture base resin should not be less than 65 MPa. The Weibull modulus reflects the strength distribution among samples; a higher Weibull modulus indicates that a larger proportion of specimens breaks within a smaller range of applied stress with a smaller range of error and higher clinical reliability. In this study, the flexural strength of the 3D-printed group after thermal cycling not only fell below the clinical application standard but also exhibited a large standard deviation in the results. Additionally, the Weibull modulus for flexural strength also decreased. These findings indicated that the quality stability of 3D-printed group deteriorated after thermal cycling. In the oral environment, the flexural strength of base resin directly affects the lifespan of dentures and the safety of patients. Consequently, it can be inferred that the 3D-printed base resin poses a higher risk of fracture or breakage in long-term clinical application and may fail to meet the masticatory function requirements of patients. The flexural strength of conventional and milled base resins showed no significant change after thermal cycling, suggesting that these two manufacturing methods can ensure greater safety for dentures during long-term use in the complex oral environment, thereby better meeting clinical requirements.

Indeed, Shore hardness and Knoop hardness are also employed to test denture base resins. However, Shore hardness is primarily used for softer materials, while Knoop hardness is predominantly applied to hard and brittle materials. In contrast, Vickers hardness offers broader applicability. According to Council on Dental Materials and Devices (1975), the Knoop hardness of PMMA base resins should not be less than 15. However, no uniform standards or specifications have been established for evaluating the Vickers hardness of base resins. Previous studies have reported that the Vickers hardness of base resin materials generally ranges between 6.68 HV and 23.9 HV, irrespective of the manufacturing process (Çakmak et al., 2023b). The results of the present study showed that the average Vickers hardness of the three groups of base resins also fell within this range (14.98 HV–20.31 HV). In this study, it was observed that the Vickers hardness of the conventional group increased significantly after thermal cycling, whereas the Vickers hardness of the milled and 3D-printed groups decreased notably after thermal cycling. The results for the conventional group were consistent with those of Al-Jmmal, Mohammed & Al-Kateb (2024). Their study showed that the Shore hardness of heat-polymerized PMMA resins increased after thermal aging (10 daily thermal cycles of the samples between 5 °C and 55 °C with a dwell time of 30 s). Ajay et al. (2020) found that residual monomers within the resin act as plasticizers, reducing polymer intermolecular forces. The thermal cycling environment in this study promoted cross-linking reactions of the resin molecules and reduced the residual monomers within the conventionally polymerized base resins, which led to an increase in Vickers hardness. Conversely, after thermal cycling, the Vickers hardness decreased in the milled and 3D-printed groups. This result may be related to the generation and accumulation of internal stresses due to the inhomogeneity of the expansion and contraction of the base resins (Al-Dulaijan et al., 2022). Internal stress changes can lead to a decrease in the structural integrity of the specimen, making it more prone to indentation. However, a study by Çakmak et al. (2023a) showed insignificant changes in Vickers hardness of 3D-printed base resins after thermal cycling. The difference from this study lies in the fact that the research involved only 5,000 thermal cycles. The limited number of cycling temperature changes may not have allowed the base resin to accumulate enough thermal stress to cause significant changes in Vickers hardness. The hardness of base resins is associated with wear resistance. Thermal cycling decreased the Vickers hardness of 3D-printed and milled base resins, while the Vickers hardness of conventional resins increased. Consequently, conventional resins better resist surface damage from temperature fluctuations in the oral environment, resulting in a longer lifespan.

Given that notching can induce stress concentration in denture base resins (Al-Dwairi, Al Haj Ebrahim & Baba, 2023), the unnotched Charpy impact strength test was selected for this study. Previous studies have shown that milled base resins exhibit superior mechanical durability, with less residual monomer (Ayman, 2017) and lower porosity (El Samahy et al., 2023), compared to both conventional and printed groups. These findings are consistent with the results of the present study, which demonstrated that the milled group maintained relatively high impact strength both before and after thermal cycling. Furthermore, this study revealed that thermal cycling significantly reduced the impact strength of the 3D-printed group, but had no significant effect on the conventional and milled groups. This may be attributed to the presence of defects, such as microcracks and fissures, within the 3D-printed resin. Robinson & McCabe (1993) demonstrated that even defects as small as 16 μm can lead to a significant reduction in the impact strength of PMMA denture base resins. These defects within the denture base resin samples, acting as stress concentration points, can easily become initiation sites for fractures during impact tests, potentially leading to a significant decrease in impact strength. However, Altarazi et al. (2024) found that the impact strength of 3D-printed base resins did not change significantly after aging. The discrepancy with the results of this study may be attributed to the fact that their research only simulated the liquid environment at oral temperature, without considering the adverse effects that temperature changes might cause. It should be noted that, compared to the conventional and milled groups, the impact strength of the 3D-printed base resin was lower both before and after thermal cycling, with a significant reduction in Weibull modulus following the cycling process. Future studies should consider improving its impact strength by optimizing printing angles, layer thickness (Alqutaibi et al., 2025), and post-curing procedure (Perea-Lowery et al., 2021) to better meet clinical requirements. The impact strength of conventional and milled base resins showed no significant change after thermal cycling, indicating their ability to withstand sudden occlusal forces or external impacts during long-term use in the oral environment. This characteristic effectively reduces the risk of damage from impacts. Therefore, for patients with strong chewing ability or high occlusal forces, clinicians may prioritize the use of conventional or milled base resins.

Both flexural strength and impact strength are indicators of a material’s toughness and are influenced by its composition and internal structure. de Jager et al. (2021) found that the impact strength of two light-cured composite resins and three universal composite resins was proportional to the flexural strength divided by the square of the flexural modulus. From this, it can be inferred that there may be a correlation between the flexural strength and impact strength of denture base resins, although this relationship is not necessarily linear and may be influenced by various factors, such as the testing method. Thomaidis et al. (2013) found no correlation between the flexural strength and impact strength of microhybrid composites, nanofilled composites, nanohybrid composites, and resin ceramic composites. This study also reported similar findings. In this study, Spearman’s correlation analysis revealed no correlation between the flexural strength and impact strength of the three groups before and after thermal cycling. This discrepancy may be due to the different loading conditions in the tests. The three-point flexural test applies gradual forces, allowing molecular chains in the base resin to align and resist the load effectively. In contrast, during the impact strength test, the base resin experiences a sudden impact load, which may hinder its ability to absorb shock energy. Furthermore, differences in sample size and shape between the two tests could also influence the results (Faot et al., 2006). This indicates that, although both flexural strength and impact strength are critical indicators of a material’s ability to resist deformation and destruction under external forces, they have distinct meanings and application focuses. When the denture base experiences frequent warping, sinking, and other unstable movements, flexural strength may be a more critical parameter. Conversely, when the denture encounters instantaneous impacts, such as collisions or falls, impact strength becomes more important. Therefore, when selecting a base resin, its mechanical properties should be considered in the context of the specific application scenario and requirements.

The composition of artificial saliva is complex, with substances such as salivary enzymes prone to interacting with the denture base resin during thermal cycling (Zappini, Kammann & Wachter, 2003). To minimize influencing factors and ensure the accuracy of the experimental results, distilled water was selected for the experiments in this study. The results of this study indicate that clinicians should consider the differences in mechanical properties among conventionally polymerized, milled, and 3D-printed base resins for intraoral use when selecting materials. Additionally, optimizing 3D printing technology is a key direction for future research to enhance its application in dental restorations.

This study still has some limitations, Only one brand of material was selected for each manufacturing technology, future research should incorporate a wider range of brands. Moreover, the three-point flexural test only assesses the static mechanical properties of the material, without simulating the dynamic changes that occur during chewing. Therefore, future studies should incorporate chewing simulators, mechanical brushing, and other devices to adequately simulate the oral environment, thereby providing more data and insights for the long-term clinical application of 3D-printed denture base resins.

Conclusions

Under the conditions of this experiment, the thermal cycling environment had a more significant effect on the flexural strength, Vickers hardness, and impact strength of the 3D-printed group compared to the conventional or milled resin materials. These findings suggest that 3D-printed base resins require further improvements (e.g., material composition, printing parameters and post-processing) in their mechanical properties before they can be applied clinically.

Supplemental Information

Supplemental Information 1 Raw data.

The raw data shows the flexural strength, Vickers hardness and impact strength of conventional, milled and 3D printed groups.

Supplemental Information 2 Temperature exchanger.

Supplemental Information 3 SEM machine.

The authors specially thank Sino-Dentex Co. Ltd for kindly supplying the base resin materials used in this experiment and Chongqing Jingmei Medical Technology Co., Ltd. for providing the samples processing platform as well as the related equipment and technical support. Special thanks to those direct and indirectly involved in the research.

Additional Information and Declarations

Competing Interests

The authors declare that they have no competing interests.

Author Contributions

Shuang Xiao performed the experiments, analyzed the data, prepared figures and/or tables, and approved the final draft.

Ruo-Jin Zhang analyzed the data, prepared figures and/or tables, and approved the final draft.

Fa-Bing Tan conceived and designed the experiments, authored or reviewed drafts of the article, and approved the final draft.

Data Availability

The following information was supplied regarding data availability:

The raw measurements are available in the Supplemental File.

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
