# Peer review of "Effect of thermal cycling on the mechanical properties of conventional, milled, and 3D-printed base resin materials: a comparative in vitro study"

_PeerJ, doi:10.7717/peerj.19141_

## Round 0.1 · original submission · Minor Revisions

Dear Authors,

Based on the reviewers' feedback, I kindly ask you to carefully address their recommendations to improve the clarity, depth, and scientific rigor of your manuscript. Please ensure that all suggested revisions are incorporated, including enhancements to the discussion, methodology details, statistical justifications, and clinical relevance. A detailed response letter outlining how each comment was addressed should be submitted along with your revised manuscript. I appreciate your efforts and look forward to receiving your revision.

Best regards,
Carlos Mourão

·

Basic reporting

1. Clear, professional English language used.
2. Introduction & background relevant
3. Figures and tables are well described.

Experimental design

4. All underlying data have been provided; statistically sound, &controlled
5. Research question well defined, relevant & meaningful.

Validity of the findings

6. Conclusions are well stated, linked to original research question & limited to supporting results.

Reviewer 2 ·

Basic reporting

-Abstract: The purpose of the study should be clearly stated. To emphasize this objective, start with "The purpose of this study is…”
- Introduction: Details about the purpose of the study in lines 60-66 should be expanded and expressed more clearly. The importance of the research and how it fills a gap in the literature should be well-articulated.
- Language and Format: The language is generally clear, but some expressions could be more professionally polished. The citation format for SPSS is incorrect. The official format should be used (e.g., "SPSS Statistics for Windows, Version 27.0. Armonk, NY: IBM Corp.").

Experimental design

- Materials and Methods:
The protocols for sample preparation in lines 69-72 should be explained in greater detail and clarity.
- In line 115, the material details should include the manufacturer name and location (e.g., "IBEE300, UNIZ Technology LLC, Beijing, China"). This format should be consistently applied to all materials throughout the manuscript.
- The rationale for choosing 1000× magnification (lines 147-177) should be clearly explained under the SEM observation section.

Validity of the findings

- Results:
- The conclusion section can be shortened. Data already presented in tables or figures should not be reiterated in detail.
- Discussion:
The discussion section is generally confusing. It should be rewritten more systematically, and more effectively comparing the study’s results with the literature.
- Limitations of the study should be explicitly mentioned.
- Statistical Analysis:
- While statistical methods are described, the manuscript does not explain why nonparametric tests were used for certain comparisons. The authors should clarify whether the data violated assumptions for parametric tests.
The correlation analysis results (Table 5) show weak and non-significant relationships. The authors could explore this further and discuss whether the lack of correlation aligns with other studies.
-SEM Observations: The discussion of SEM images (Figure 6) is descriptive but lacks quantitative insights. Adding measurements (e.g., crack width, and defect sizes) would strengthen the analysis.

Additional comments

Information such as manufacturer details and LOT numbers for the materials used is missing from Table 1 and should be added.
- The definitions of uppercase and lowercase letters in Table 2 are difficult to understand. They should be expressed more clearly.
- Figure 4 shows Vickers microhardness images. Is this figure necessary? It may be more appropriate to remove it from the manuscript.
-While the study concludes that 3D-printed materials need improvement, the authors could elaborate more on specific strategies for enhancing these materials, such as optimizing polymer compositions or adjusting printing parameters (e.g., layer thickness).
- The discussion does not address the environmental impact of the different fabrication methods (conventional, milled, and 3D-printed). Given the rising importance of sustainable practices in dentistry, this could be a valuable addition.

Reviewer 3 ·

Basic reporting

The literature review, while thorough, could better elaborate on the clinical implications of the observed mechanical properties.

Experimental design

no comment

Validity of the findings

The conclusion statement seems too brief. Please suggest some improvement strategies for the mechanical properties of the 3D-printed group samples to meet clinical applications.

Additional comments

The results show that the 3D-printed denture resin materials do not perform as well as conventional denture resin after thermal cycling. To confirm whether the issues are caused by 3D printing or the material composition, you can do a comparative experiment by curing the exact same formulation in a mold and compare the mechanical properties with conventional denture materials.

Reviewer 4 ·

Basic reporting

This study is stated clearly and concisely and the literature review is sufficient.

Experimental design

Materials and methods described with sufficient detail & information.

Validity of the findings

All underlying data have been provided; they are robust, statistically sound, & controlled. Conclusions are well stated, linked to original research question & limited to supporting results.

Additional comments

Although it is a very well-designed and fluently written manuscript, I have a few suggestions.
It is sufficient to mention the oral environment and its characteristics between lines 39-59 in the introduction section. The rresearchs conducted should be written in the discussion section.
In the discussion section, the results of the study should be discussed and the clinical importance of this in vitro study or recommendations that can be given to the clinician should be mentioned.

---

## Round 0.2 · accepted · Accept

Thank you for addressing the reviewer’s comments and suggestions. I confirm that in my opinion the article is now Acceptable